# Assessment of the Polyphenolic Profile and Beneficial Effects of Red and Green Propolis in Skin Inflammatory Conditions and Oxidative Stress

**DOI:** 10.3390/biomedicines13092229

**Published:** 2025-09-10

**Authors:** Andrea Magnavacca, Giulia Martinelli, Nicole Maranta, Carola Pozzoli, Marco Fumagalli, Giangiacomo Beretta, Stefano Piazza, Mario Dell’Agli, Enrico Sangiovanni

**Affiliations:** 1Department of Pharmacological and Biomolecular Sciences “Rodolfo Paoletti”, Università degli Studi di Milano, 20133 Milan, Italy; andream93@libero.it (A.M.); giulia.martinelli@unimi.it (G.M.); nicole.maranta@unimi.it (N.M.); carola.pozzoli@unimi.it (C.P.); marco.fumagalli3@unimi.it (M.F.); mario.dellagli@unimi.it (M.D.); enrico.sangiovanni@unimi.it (E.S.); 2Department of Environmental Science and Policy, Università degli Studi di Milano, 20133 Milan, Italy; giangiacomo.beretta@unimi.it

**Keywords:** Brazilian propolis, skin, Inflammation, keratinocyte, fibroblast

## Abstract

**Background/Objectives:** Propolis is a complex natural product with long-standing traditional use as an antimicrobial remedy. Several studies suggest that Brazilian varieties of propolis may promote wound healing and protect the skin from UV damage, most likely due to antioxidant and anti-inflammatory mechanisms. However, the literature provides limited support for this topic. The present work aimed at characterizing the polyphenolic profile of two Brazilian propolis samples, investigating their biological activity. **Methods:** Biological experiments were conducted in human keratinocytes (HaCaT) and fibroblasts (HDF) stimulated by cytokines involved in skin inflammation and remodeling (TNF-α and IL-1β), while phytochemical analyses were conducted by LC-MS techniques. **Results:** Our findings indicate that artepillin C and drupanin were the principal phytochemicals of green propolis, while vestitol, medicarpin, and neovestitol were the most abundant in red propolis. The presence of phenolic compounds was correlated with the antioxidant activity demonstrated by ORAC and intracellular ROS assays. Accordingly, both Brazilian propolis samples impaired NF-κB activity, while only red propolis hindered IL-8 release in both cell lines with an IC_50_ lower than 25 μg/mL. Surprisingly, both propolis samples at the same concentrations enhanced the production of IL-6 and VEGF, thus suggesting the coexistence of anti-inflammatory, antioxidant, and trophic mechanisms contributing to skin repair. In line with this hypothesis, propolis also induced the stabilization of HIF-1α, paralleling the biological effect of a well-known synthetic HIF stabilizer (DMOG). **Conclusions:** This work supports the investigation of Brazilian red and green propolis as potential modulators of the inflammatory phase in wound healing.

## 1. Introduction

Propolis is a complex resinous mixture composed of exudates gathered by honeybees (*Apis mellifera* Linnaeus, 1758) from buds, twigs, sprouts, vegetal apices, sap flows, or other botanical sources and elaborated with beeswax and salivary secretions [1], traditionally used for the topical treatment of burns and minor injuries [2] due to its renowned antimicrobial [3], antioxidant [4], and anti-inflammatory properties [5]. In 2007, this classification was further expanded with the discovery of a thirteenth variety native to the states of Sergipe, Alagoas, Paraíba, Pernambuco, and Bahia known as red propolis [6]. The main source of type-12 green propolis, the most widespread and best-selling variety, is represented by the buds and non-expanded leaves of *Baccharis dracunculifolia* DC. belonging to the family Asteraceae [7], whereas the main botanical source of red propolis is *Dalbergia ecastaphyllum* (L.) Taub., a leguminous plant known for the presence of red pigments composed of cationic C_30_ isoflavans [8].

The chemical profile of propolis has been recently defined by the International Organization for Standardization (INTERNATIONAL STANDARD, ISO 24381:2023, Bee propolis – specifications; 2023). The most abundant components of green propolis are members of the phenylpropanoid class, such as cinnamic acid, coumaric acid, and caffeic and caffeoylquinic acids [9]. The presence of prenylated phenylpropanoids is noteworthy, in particular, 4-hydroxycinnamic acid prenylated derivatives [10], which can be distinguished between cyclized chromenes, such as 2,2-dimethyl-8-prenylchromene, and compounds that do not cyclize such as artepillin C and drupanin [11]. Artepillin C (3,5-diprenyl-4-hydroxycinnamic acid) stands out due to its abundance and important biological activities [12]. Flavonoids are minor components of green propolis, whereas they are undoubtedly predominant in red propolis [13], in particular, in the subclass of isoflavonoids. The principal molecules identified in red propolis belong to the classes of flavonols (quercetin and rutin), flavanonols (pinobanksin), flavones (luteolin), flavanones (liquiritigenin and pinocembrin-3-acetate), isoflavones (daidzein), the neoflavone dalbergin, and the chalcone isoliquiritigenin [6]. Other compounds identified include vestitol and neovestitol, formononetin, biochanin A, pinocembrin, and medicarpin, which are characteristic markers used to identify red propolis samples [14].

Propolis has demonstrated antimicrobial [3], antiviral [15], antioxidant, anti-inflammatory, and immunostimulant properties [5]. Propolis can regulate the synthesis and release of the principal pro-inflammatory cytokines, including IL-8, IL-6, IL-1β, and IL-12, while increasing the expression of anti-inflammatory cytokines such as IL-4 and IL-10 [16,17]. Many of the major anti-inflammatory mechanisms of action are related to the modulation of NF-κB and AP-1 signaling [18].

Several in vitro and in vivo studies suggest that Brazilian propolis may promote wound healing [19,20,21] and protect the skin from UV damage [22,23]. Nevertheless, the mechanisms and the putative compounds responsible for the biological effects at the skin level were poorly investigated. Besides classical inflammatory pathways, such as NF-κB, polyphenols are known to interfere with multiple pathways involved in cell metabolism, including HIF-1 [24,25].

The role of HIF-1 as a potential target in epithelial barrier defects, as is the case in skin and intestinal inflammatory diseases, has been a topic of discussion in recent research [26,27]. Inflammation and hypoxia signaling are mutually interdependent: inflammatory states are frequently characterized by tissue hypoxia, or otherwise by the stabilization of hypoxia-dependent transcription factors [28]. HIF-1 influences the cellular inflammatory response through a metabolic switch to glycolysis, which may be important for preventing the excessive generation of deleterious reactive oxygen species and impaired wound resolution [29], and elicits the upregulation of many genes that enhance the wound repair process, such as adhesion proteins, soluble growth factors (TGF-β and VEGF), and matrix components. Consequently, the anti-inflammatory effects of hypoxia signaling have been linked to a transcriptional program under the control of HIF-1, which has been shown to dampen hypoxia-induced inflammation in a wide variety of inflammatory disease models, for example, through the enhanced production of anti-inflammatory signaling molecules [30,31].

The interplay among keratinocytes and fibroblasts is fundamental in skin repair: keratinocytes play a key role in skin barrier and immunity, while fibroblasts are responsible for tissue remodeling and cell migration [32].

Within this context, a single study, limited to Brazilian green propolis, has highlighted a possible involvement in the modulation of the activity of the transcription factor HIF-1. However, available data refer to a model of human embryonic kidney (HEK) cells transfected with a reporter plasmid [33]. Drupanin, baccharin, and kaempferide inhibited the activation of HIF-1α and downstream target genes such as *GLUT1* and *VEGFA*, whereas the flavonoids isosakuranetin and betuletol induced HIF-1 transcriptional activity in both hypoxic and normoxic conditions [33].

In the present work, on the one hand, an analytical characterization of green and red Brazilian propolis samples was performed, while on the other hand, their spectrum of molecular mechanisms on skin cells was evaluated. A comprehensive in vitro assessment of the still largely unknown biological activities of green and red Brazilian propolis on human keratinocyte and dermal fibroblast cellular models was conducted, taking into account the potential involvement of NF-κB and HIF-1 pathways. In particular, the purpose of the investigation of the anti-inflammatory, HIF-1-modulating activities, and their interrelationships was intended to showcase the potential of Brazilian propolis as an all-around skin-protecting agent able to elicit a regenerative cell behavior triggered by a hypoxia-mimicking survival response.

## 2. Materials and Methods

### 2.1. Brazilian Propolis Samples

Green propolis tincture, a dark green hydroalcoholic solution certified to contain at least 11% of green propolis, was purchased from “Apiário Silvestre” (Piracaia, Sâo Paulo State, Brazil). Red propolis tincture, a carmine, aromatic hydroalcoholic solution, was purchased from “Apiário Cajueiro” (Una, Bahia State, Brazil). Both propolis tinctures were kept overnight at −20 °C to favor the precipitation of inert waxes, which were then filtered out by vacuum filtration. The filtrate was evaporated to dryness using a rotary evaporator (Laborota 4000 efficient, Heidolph Instruments GmbH & Co., Schwabach, Germany), and the dry residue was redissolved at a concentration of 50 mg/mL in a mixture of 75:25 EtOH:H_2_O, aliquoted, and stored at −20 °C for subsequent experiments.

### 2.2. Total Phenolic Content (TPC)

The total phenolic content was determined according to the Folin–Ciocâlteu method. Briefly, propolis samples were preliminarily diluted in deionized water (1 mg/mL), and 20 µL, corresponding to 20 μg of extract, was further diluted to a final volume of 800 µL. Then, 50 µL of 2 N Folin–Ciocâlteu reagent (Merck Life Science, Milan, Italy) and 150 µL of 20%_w/v_ Na_2_CO_3_ were added. After 30 min of incubation at 37 °C, the absorbance of the samples was measured with a JASCO V630 UV-Vis cuvette spectrophotometer (JASCO International Co. Ltd., Tokyo, Japan) at 765 nm. The total phenolic content was quantified using a calibration curve of gallic acid (0–15 μg/mL).

### 2.3. Oxygen Radical Absorbance Capacity (ORAC) Assay

The oxygen radical absorbance capacity assay was carried out according to Dávalos et al. [34], with minor modifications. Briefly, propolis samples were preliminarily diluted in deionized water (at 5 μg/mL), and 20 µL, corresponding to 0.1 μg of extract, was aliquoted into a black 96-well plate (Greiner Bio-One Italia S.r.l., Cassina de’ Pecchi, Italy). Then, 120 µL of fluorescein solution (with a final concentration of 70 nM) in a phosphate buffer (75 mM, pH 7.4) was added to each well. Alkyl peroxyl radicals were generated by the addition of 60 µL of 40 mM AAPH solution [2-2′-azobis(2-aminidinopropane) dihydrochloride] (Merck Life Science, Milan, Italy). The plate was immediately put in a multilabel plate reader (VICTOR X3, PerkinElmer Italia S.p.a., Milan, Italy) at 37 °C, and fluorescence (Ex. = 485 nm/Em. = 535 nm) was recorded, after shaking, every 2 min for 31 repeats. Trolox (0–120 µM) was used as a reference radical scavenger. The area under the curve (AUC) was calculated for each sample and compared with the standard Trolox curve.

### 2.4. GC-MS Analysis

A few milligrams of propolis extract, evaporated to dryness, were treated with 70 µL of the silylating agent BSTFA [N,O-bis(trimethylsilyl)trifluoroacetamide] (Merck Life Science, Milan, Italy) in 30 µL of pyridine and 100 µL of ethyl acetate. The reaction was conducted for 3 h at 70 °C, and then 1.0 µL was injected into the gas chromatograph. GC-MS analysis was carried out on a Bruker SCION SQ gas chromatograph (Bruker Daltonics, Macerata, Italy) equipped with a Zebron ZB-5HT Inferno column (30 m; ID 0.25 mm, df 0.25 µm) (Phenomenex, Castel Maggiore, Italy) coupled with an EI source and single quadrupole (SQ) analyzer.

Operative parameters were set as follows: the GC oven temperature program was 60 °C (3 min), 8.0 °C/min to 120 °C (1 min hold), 4.0 °C/min to 280 °C (1.5 min hold), and 10.0 °C/min to 380 °C (2 min hold) for a total runtime of 60 min; the inlet temperature was 250 °C; the flow rate was 1.00 mL/min; the carrier gas was helium 5.5; and the ionization energy was 70 eV. The split/splitless ratio was 1:30 after 45 s. Peaks were assigned by matching experimental mass spectra with those present in the NIST mass spectral library (version 2.0, 2011). The relative abundance of compounds was obtained through peak area normalization.

### 2.5. HPLC-ESI-HRMS Analysis

Analyses were conducted on ethanolic samples, further diluted to 1:20 in ethanol, and centrifuged (16,000× *g*, 5 min) using a SYNAPT G2-Si mass spectrometer (Waters, Sesto San Giovanni, Italy) equipped with an autosampler, a UPLC-PDA detector operating in the range of λ = 190–400 nm, an ESI source, and a TOF analyzer. The chromatographic separation was obtained using an ACQUITY UPLC HSS T3 column (75 mm × 2.1 mm, 1.8 μm) (Waters, Sesto San Giovanni, Italy). The injection volume was 10.0 µL; the mobile phase was set as follows: acetonitrile (mobile phase A) and water (mobile phase B) with a gradient of 10% A for 2 min, then 10–60% in 48 min, and then 60–90% in 10 min. Desolvation was obtained with nitrogen heated at 180 °C and a capillary voltage of 2 kV. Data was acquired in a negative mode.

### 2.6. Cell Cultures

HaCaT cells (RRID:CVCL-0038, Cell Line Service GmbH, Eppelheim, Germany), a stable cell line of human keratinocytes from Caucasian male adults, were grown as an adherent monolayer in a high-glucose DMEM medium (Merck Life Science, Milan, Italy) supplemented with penicillin (100 units/mL) and streptomycin (100 μg/mL) (Pen Strep Gibco, Thermo Fisher Scientific, Rodano, Italy), 2 mM L-glutamine (Gibco, Thermo Fisher Scientific, Rodano, Italy), and 10% heat-inactivated fetal bovine serum (Euroclone S.p.A, Pero, Italy), at 37 °C in a humidified atmosphere with 5% CO_2_.

HDF cells (RRID:CVCL_UF42, ECACC, Porton Down, Porton, UK), a stable cell line of normal female adult dermal fibroblasts, were grown as an adherent monolayer in a DMEM medium (Merck Life Science, Milan, Italy) supplemented with penicillin (100 units/mL) and streptomycin (100 μg/mL) (Pen Strep Gibco, Thermo Fisher Scientific, Rodano, Italy), 2 mM L-glutamine (Gibco, Thermo Fisher Scientific, Rodano, Italy), and 10% heat-inactivated fetal bovine serum (Euroclone S.p.A, Pero, Italy), at 37 °C in a humidified atmosphere with 5% CO_2_.

At 80–90% confluence, cells were detached from 75 cm^2^ culture flasks (Euroclone S.p.A, Pero, Italy) using a 0.25% trypsin–EDTA solution (Gibco, Thermo Fisher Scientific, Rodano, Italy) and transferred to a new flask at a density of 1.5 × 10^6^ cells per flask to allow for cell line expansion. For the experimental procedures, cells were seeded in flat-bottom culture plates or dishes (Falcon, Corning Life Sciences B.V., Amsterdam, The Netherlands) at a standard density of 1.5–3 × 10^4^ cells/cm^2^ and cultured for 72 h before the treatment.

### 2.7. Cell Treatments

After 72 h of growth, HaCaT and HDF cells were treated with different concentrations of green and red propolis extracts using a serum-free medium. Concomitantly, cells were stimulated with the pro-inflammatory cytokines TNF-α or IL-1β (10 ng/mL) for 6 h or 24 h. Moreover, according to previously published methodologies [35], IL-6 production was evaluated by adding the inflammatory enhancer IFN-γ (5 ng/mL) to TNF-α. Following previous works, 20 μM of epigallocatechin gallate (EGCG) was used as a reference inhibitor against TNF-α, while 1 mM of DMOG was used as a reference inhibitor against IL-1β.

### 2.8. Cytotoxicity Assays

The integrity of cell morphology before and after the treatments was assessed by light microscope inspection. The viability of HaCaT and HDF cells in the presence of propolis samples was investigated by MTT [3-(4,5-Dimethyl-2-thiazolyl)-2,5-diphenyl-2H-tetrazolium bromide] and LDH (lactate dehydrogenase) release tests (Smith, Wunder, Norris, & Shellman, 2011) [35].

For the MTT test, the medium was removed after treatment and replaced with a 0.25 mg/mL MTT (Merck Life Science, Milan, Italy) solution in PBS; then, the cells were incubated for 30 min at 37 °C. At the end of incubation, the MTT solution was discarded, formazan crystals were dissolved with a 90:10 isopropanol:DMSO solution, and the absorbance was measured at 570 nm using a multilabel plate reader (EnVision 2101, PerkinElmer Italia S.p.a., Milan, Italy).

For the LDH assay, the amount of LDH released in the medium compared to total intracellular LDH was measured with the LDH Cytotoxicity Detection Kit (Takara Bio Europe, Saint Germain-en-Laye, France), following the manufacturer’s instructions. After 30 min of incubation at room temperature, absorbance was read at 490 nm using a multilabel plate reader (VICTOR X3, Perkin Elmer S.p.a., Milan, Italy).

### 2.9. Reporter Plasmid Assays

Two reporter plasmids were used to transiently transfect HaCaT and HDF cells, as previously described [35]: NF-κB-Luc, a luciferase reporter construct containing three κΒ-responsive elements from the E-selectin gene, was a gift from Dr N. Marx (Department of Internal Medicine-Cardiology, University of Ulm, Ulm, Germany); HRE-Luc, a luciferase reporter construct containing three hypoxia-responsive elements (24-mers) from the *Pgk-1* gene, was a gift from Navdeep Chandel (Addgene plasmid # 26731; http://n2t.net/addgene:26731, accessed on 7 January 2025; RRID:Addgene_26731).

Plasmid amplification was obtained in transformed *Escherichia coli*, strain DH5α. Bacteria from frozen glycerinates were grown in LB broth supplemented with 100 μg/mL ampicillin (Merck Life Science, Milan, Italy), keeping flasks overnight in an orbital shaker at 37 °C. The following day, bacteria were pelleted by centrifugation (5000 rpm, 10 min, 4 °C), and the plasmid was extracted using a commercial kit according to the manufacturer’s instructions (NucleoBond^®^ Xtra Maxi; MACHEREY-NAGEL GmbH & Co. KG, Düren, Germany). The purified plasmid was resuspended in nuclease-free water and quantified with a NanoDrop ND-1000 spectrophotometer (Thermo Fisher Scientific, Rodano, Italy).

After preliminary titration of reporter plasmid signal-to-noise response, the amounts of 250 ng/well for 24-well plates and 42 ng/well for 96-well plates were chosen. Cells were transiently transfected with Lipofectamine 3000 transfection reagent (Thermo Fisher Scientific, Rodano, Italy). After incubation overnight, cells were treated, and the luciferase produced was assessed using the Britelite Plus reporter gene assay system (PerkinElmer Italia S.p.a., Milan, Italy), according to the manufacturer’s instructions. The luminescence generated by the reaction between luciferase and luciferin was measured with a multilabel plate reader (VICTOR X3, PerkinElmer Italia S.p.a., Milan, Italy).

### 2.10. ELISA

The release of IL-8, IL-6, and VEGF was evaluated by an enzyme-linked immunosorbent assay (ELISA) on culture media [35]. Human IL-8 (ABTS), IL-6 (TMB), and VEGF (ABTS) ELISA development kits were purchased from PeproTech (PeproTech, London, UK). In brief, 96-well EIA/RIA plates (Corning Life Sciences B.V., Amsterdam, The Netherlands) were coated overnight at room temperature with a capture antibody contained in the kit. The amounts of IL-8, IL-6, and VEGF in the samples were detected by measuring the absorbance resulting from the colorimetric reaction between an HRP conjugate and 2,2′-azino-bis(3-ethylbenzothiazoline-6-sulfonic acid) (ABTS) or a 3,3′,5,5′-tetramethylbenzidine (TMB) substrate (Merck Life Science, Milan, Italy). Absorbance was measured at 405 nm (ABTS) or 450 nm (TMB) using a multilabel plate reader (VICTOR X3; PerkinElmer, Milan, Italy). IL-8, IL-6, and VEGF levels were extrapolated from a standard curve of the mediator of interest.

### 2.11. Immunocytochemistry

For immunocytochemistry experiments, cells were seeded on Nunc Lab-Tek II chamber slides (Thermo Fisher Scientific, Rodano, Italy). At the end of the treatment, cells were washed with PBS and fixed with buffered 4% formaldehyde solution for 15 min. Cells were washed 3 times with PBS and incubated for 1 h in a blocking solution consisting of 5% normal goat serum (Cell Signaling Technology B.V., Leiden, The Netherlands) and 0.3% Triton X-100 (Merck Life Science, Milan, Italy) in PBS. Then, the blocking solution was removed, and cells were incubated overnight at 4 °C with the primary antibody (HIF-1α D1S7W XP Rabbit mAb #36169, Cell Signaling Technology B.V., Leiden, The Netherlands) diluted to 1:800 in 1% BSA (Merck Life Science, Milan, Italy) and 0.3% Triton X-100 (Merck Life Science, Milan, Italy) in PBS. The day after, cells were washed 3 times with PBS and incubated for 1.5 h with the secondary antibody (Anti-rabbit IgG (H+L) F(ab’)_2_ Fragment Alexa Fluor 647 conjugate #4414, Cell Signaling Technology B.V., Leiden, The Netherlands) diluted to 1:1500 and ActinRed 555 (Thermo Fisher Scientific, Rodano, Italy). Finally, cells were washed 3 times with PBS, mounted with coverslips using a drop of ProLong Gold antifade reagent with DAPI (Cell Signaling Technology B.V., Leiden, The Netherlands), and imaged with a confocal laser scanning microscope (LSM 900, Carl Zeiss S.p.a., Milan, Italy).

### 2.12. Gene Expression Analysis

RNA was extracted at the end of the treatment, the medium was removed, and cells were washed with PBS and lysed with QIAzol Lysis Reagent (QIAGEN S.r.l., Milan, Italy) according to the manufacturer’s instructions. The lysates were homogenized and stored at −80 °C until the following RNA purification steps were carried out: Total RNA was isolated from cell lysates with miRNeasy Mini Kit (QIAGEN S.r.l., Milan, Italy), according to the manufacturer’s protocol. A set of RNase-free DNase (QIAGEN S.r.l., Milan, Italy) was used to provide efficient on-column digestion of genomic DNA. Total RNA was eluted with nuclease-free water, and the concentration and quality were assessed spectrophotometrically using a NanoDrop ND-1000 spectrophotometer (Thermo Fisher Scientific, Rodano, Italy). Sample purity was estimated by measuring A260/280 and A260/230 ratios to check for possible contaminants co-purified during the RNA isolation.

Then, real-time quantitative PCR was performed using the iScript One-Step RT-PCR kit for probes (Bio-Rad Laboratories S.r.l., Segrate, Italy) according to the manufacturer’s instructions, on a CFX38 Real-Time PCR Detection System (Bio-Rad Laboratories S.r.l., Segrate, Italy). Primers and probes were obtained from Eurofins Genomics Italy (Vimodrone, Italy) (Table 1).

The threshold cycle value for each gene was automatically provided by the software CFX Manager (Bio-Rad Laboratories S.r.l., Segrate, Italy), depending on the amplification curves, and quantification was performed with the standard curve method. Expression data were normalized to the housekeeping gene *36B4* (ribosomal protein lateral stalk subunit P0).

### 2.13. Intracellular ROS Production

ROS production was investigated in HaCaT cells seeded in 96-well black plates, as previously described [36]. At the end of 24 h pretreatment with propolis extracts, cells were washed with warm PBS and incubated at 37 °C with 25 μM CM-H2DCFDA (Invitrogen, Thermo Fisher Scientific, Monza, Italy) in HBSS without phenol red (Merck Life Science, Milano, Italy). After 30 min, cells were washed twice with PBS and treated for 1 h with H_2_O_2_ (1 mM) in serum-free medium. Following 1 h of incubation, cells were washed again with PBS, and 100 μL of PBS was added to each well; finally, fluorescence was read using a multiplate reader (VICTOR X3; PerkinElmer, Milano, Italy) at Ex. = 490 nm/Em. = 535 nm.

### 2.14. Statistical Analysis

All the experiments were performed at least in triplicate. The results of TPC and ORAC assays are expressed as mean ± SD for unpaired two-sample *t*-tests. All other results are expressed as mean ± SEM, and the statistical significance of differences between means has been calculated through an unpaired one-way ANOVA followed by a Bonferroni post hoc test for multiple comparisons. Statistical analyses and IC_50_ calculations were carried out using GraphPad Prism 8.0.1 (GraphPad Software Inc., San Diego, CA, USA); *p* values less than 0.05 were considered statistically significant.

## 3. Results

### 3.1. Total Phenolic Content and Antioxidant Capacity of Brazilian Propolis Extracts

In the first step of this work, two propolis tinctures from the Brazilian market were refined by removing waxes through simple procedures described in Methods. Then, dry extracts were characterized for the total phenolic content (TPC) and oxygen radical absorbance capacity (ORAC).

The TPC of Brazilian green and red propolis dry extracts was determined using the Folin–Ciocâlteu assay. Table 2 shows the TPC values of propolis dry extracts and their relationship with the original hydroethanolic tinctures.

A statistically significant difference in TPC values was observed between the two propolis samples, with red propolis showing a TPC approximately twice that of green propolis.

The two propolis samples demonstrate a statistically significant difference in the antioxidant capacity (Table 3).

Considering the potential involvement of phenolic compounds in determining the antioxidant capacity, red propolis shows an antioxidant capacity that is approximately twice that of green propolis, which is consistent with the results of the TPC.

### 3.2. GC-EI-MS

The GC-EI-MS chromatograms of silylated green and red Brazilian propolis samples are shown in Figure 1. Even in this case, it is evident that the two propolis samples possess completely different composition profiles.

The putative assignment of peaks was conducted by comparison of recorded mass spectra with those in the NIST 2011 mass spectral library or those reported in the literature [37,38,39]. Regarding green propolis, peak numbers reported in Figure 1A were related to the compounds listed in Table 4.

In green propolis, the relative intensity of peaks demonstrates that the main compounds, identified as TMS derivatives, are artepillin C (*m*/*z* = 444.3 [M+H]^+^, peak 9), drupanin (*m*/*z* = 376.2 [M+H]^+^, peak 8), *p*-coumaric acid (*m*/*z* = 308.1 [M+H]^+^, peak 4), and some sesquiterpenes such as cycloartenol acetate (*m*/*z* = 468.5 [M+H]^+^, peak 12) and β-amyrin (*m*/*z* = 498.5 [M+H]^+^, peak 10), the structures of which are shown in Figure 2.

Regarding red propolis, peak numbers reported in Figure 1B were related to the compounds listed in Table 5.

In red propolis, the relative intensity of peaks demonstrates that the principal compounds, identified as TMS derivatives, are vestitol (*m*/*z* = 416.2 [M+H]^+^, peak 11), neovestitol (*m*/*z* = 416.2 [M+H]^+^, peak 10), and medicarpin (*m*/*z* = 342.2 [M+H]^+^, peak 9), while liquiritigenin (*m*/*z* = 400.2 [M+H]^+^, peak 12) and formononetin (*m*/*z* = 340.2 [M+H]^+^, peak 13) are marginally present. Based on these results, it was hypothesized that neovestitol and vestitol represented the two main compounds observed in GC-EI-MS analysis, the structures of which, together with those of other identified compounds, are reported in Figure 3.

### 3.3. HPLC-ESI-HRMS

The investigation into the phytochemical composition was concluded by a thorough comparative HPLC-UV/PDA-ESI-HRMS analysis, which is reported in the Appendix A (Appendix A).

All compounds detected in green propolis are reported in Table 6: the main identified compounds were *p*-coumaric acid (*m*/*z* 163.0399 [M-H]^−^, peak 1), drupanin (*m*/*z* 231.1029 [M-H]^−^, peak 4), and artepillin C (*m*/*z* 299.1633 [M-H]^−^, peak 8); the structures of the respective molecules are reported in Appendix A.

The compounds identified in red propolis are reported in Table 7: the main compounds identified were vestitol and neovestitol (*m*/*z* 271.0974 and *m*/*z* 271.0972 [M-H]^−^, peaks 3 and 4, respectively), medicarpin/4′-dihydroxy-2-methoxychalcone/(7 S)-dalbergiphenol (all with *m*/*z* 269 and the same molecular formula, peak 5), and guttiferone (*m*/*z* 601.3535 and *m*/*z* 601.3531 [M-H]^−^, peaks 6 and 7, respectively). The structures of the identified molecules are reported in Appendix A.

The assignment of the main peaks to the respective compounds was made possible by the comparison of spectroscopic and spectrophotometric parameters with those reported in the literature [green propolis: [37,40]; red propolis: [37,41]. Since the signal was acquired in a negative ion mode, MS peak assignment was conducted taking into consideration *m*/*z* values corresponding to [M-H]^−^. Further confirmation was obtained through the elemental analysis of *m*/*z* fragments of interest.

In some cases, besides the deprotonated pseudomolecular ion peaks, it was possible to appreciate ion peaks with masses corresponding to [2M-H]^−^, indicating molecule dimerization within the ion source. In Appendix A, the case of neovestitol/vestitol is reported as an example.

### 3.4. In Vitro Cytotoxicity

Before proceeding with the evaluation of the biological activities in keratinocytes and fibroblasts, cytotoxicity was evaluated to assess the influence on cell viability and determine the maximum concentrations to be used in cell treatments.

At 6 h, green Brazilian propolis extract did not affect viability, neither in HaCaT nor HDF cells (Appendix A). On the other hand, red propolis extract induced a slight reduction in viability in HaCaT cells, which was measured by the MTT assay at the concentrations of 25 and 50 μg/mL. Given a certain variability and to dispel possible doubts, cytotoxicity at the highest concentrations was assessed by LDH assay as well, confirming the absence of significant cytotoxic effects. Consequently, in 6 h treatments, both propolis samples were used up to a concentration of 50 μg/mL.

At 24 h, once again, green propolis extract did not affect viability, either in HaCaT or HDF cells (Appendix A). Instead, red propolis extract showed significant cytotoxic effects in HaCaT cells at the concentrations that had yielded dubious results at 6 h. Therefore, in 24 h treatments in HaCaT cells, red propolis was used up to a concentration of 10 μg/mL.

### 3.5. Effect on NF-κB-Driven Transcription

As the very first step in the biological investigation, propolis was screened against a key pathway of skin inflammation, also known as the NF-κB pathway. Both TNF-α and IL-1β are inflammatory cytokines involved in skin diseases associated with tissue damage. However, they are known to activate the NF-κB pathway in diverse ways. Thus, both cytokines were selected for our experimental setting to obtain more information regarding anti-inflammatory mechanisms occurring at the upstream level of NF-κB activation.

For this purpose, cells were transiently transfected with an NF-κB-Luc reporter plasmid, subjected to the pro-inflammatory stimulus, TNF-α or IL-1β (10 ng/mL), and concomitantly treated with increasing concentrations of Brazilian propolis (1–50 μg/mL).

Beginning with the results obtained with TNF-α stimulation of HaCaT cells (Figure 4A,B), green propolis was able to partially inhibit NF-κB-driven transcription, in a statistically significant manner, only at the concentrations of 25 and 50 μg/mL, with an IC_50_ of 40.29 μg/mL. The effect of red propolis is greater and markedly concentration-dependent, with an IC_50_ of 8.51 μg/mL. At 25 and 50 μg/mL, NF-κB-driven transcription was brought back to basal levels. In HDF cells (Figure 4C,D), in the case of green propolis, a concentration-dependent response was not observed, even though, at the highest concentrations, a slight inhibition of NF-κB-driven transcription could be seen. On the contrary, despite the entity of the effect being lower compared to HaCaT cells, red propolis was once again able to bring NF-κB-driven transcription back to basal levels in a concentration-dependent manner, with an IC_50_ of 23.83 μg/mL.

Stimulation with IL-1β was investigated only in HaCaT cells, since HDF cells were not elicited by this cytokine. Green propolis determined a slight inhibition of IL-1β-induced NF-κB-driven transcription at the highest concentration tested of 50 μg/mL (Figure 5A). The effect of red propolis was greater, with an IC_50_ of about 10 μg/mL (Figure 5B). At the concentrations of 25 and 50 μg/mL, NF-κB-driven transcription was brought back to basal levels.

### 3.6. Effect on IL-8 and IL-6 Secretion

Since Brazilian propolis showed an inhibitory effect on NF-κB activity, we selected various soluble pro-inflammatory mediators as read-outs of such an inhibitory effect at a downstream level. Thus, the release of NF-κB-dependent mediators, namely IL-8 and IL-6, was measured by ELISA.

Following the previous experimental paradigm, to assess the ability of green and red Brazilian propolis to inhibit IL-8 release in keratinocytes and fibroblasts, cells were subjected to a pro-inflammatory stimulus (TNF-α 10 ng/mL, 6 h) and concomitantly treated with increasing concentrations of propolis (1–50 μg/mL). In HaCaT cells (Figure 6A,B), green propolis did not prove to be appreciably active in inhibiting TNF-α-induced IL-8 release. Red propolis, instead, showed statistically significant activity at concentrations higher than 10 μg/mL, with an IC_50_ of 11.89 μg/mL.

In HDF cells (Figure 6C,D), green propolis demonstrated a limited inhibitory activity on TNF-α-induced IL-8 release, with a slight and non-statistically significant reduction at 25 μg/mL. In the same experimental conditions, red propolis showed significant concentration-dependent inhibition of IL-8 release starting at 1 μg/mL, with an IC_50_ of 5.89 μg/mL.

The release of IL-6 was also investigated under a paradigm published in previous studies [35], challenging cells with a combination of TNF-α (10 ng/mL) and IFN-γ (5 ng/mL).

Unexpectedly, in HaCaT cells (Figure 7A,B), the highest concentration tested, which was the same used in the previous experiments, caused a significant increase in IL-6 release above the stimulated levels. In HDF cells (Figure 7C,D), the direction of the effect was similar to HaCaT cells but of modest magnitude.

Notably, in both HaCaT (Figure 8A) and HDF (Figure 8B) cells, green propolis and red propolis were able to significantly induce the overexpression of the *IL6* gene, thus confirming the data reported in Figure 7. The effect induced by red propolis was significantly higher than that of green propolis.

IL-6 exerts pleiotropic functions in inflammation, and it is also involved in complex events related to tissue remodeling and fibrosis, not only at the skin level. It generally demonstrates its effects in the initial phases of wound healing, making it favorable for a successful outcome: in fact, IL-6 is a key modulator of the inflammatory and reparative process involved in the differentiation, activation, and proliferation of leukocytes, endothelial cells, keratinocytes, and fibroblasts [42]. Moreover, it is also crucial for the formation of new vessels and collagen deposition. Notably, IL-6 knockout mice show a delay in wound healing [43].

Based on these observations, at this step of our work, we wondered whether Brazilian propolis might sustain the production of other inflammatory markers involved in tissue remodeling, such as VEGF. Thus, the expression of *VEGFA* was investigated by qPCR in HaCaT and HDF cells to parallel previous data regarding *IL6*.

In both cell lines (Figure 8C,D), Brazilian propolis extracts induced a significant overexpression of *VEGFA*, thus sustaining our initial hypothesis. However, results differed from those obtained for *IL6*, since the acme of *VEGF* induction in HaCaT cells was detected after 24 h of treatment with green propolis.

Moreover, the amount of VEGF released in the culture medium was measured at 24 h by ELISA to determine whether the secretion paralleled the overexpression detected by qPCR. In HaCaT cells (Figure 9), only the highest concentration tested, which was the same as that used in the previous experiments, caused a significant increase in VEGF release. In HDF, unexpectedly, no secretion could be detected.

The modulation of inflammatory markers determined by propolis extracts drew our attention to other biological mechanisms relevant to tissue remodeling, such as the modulation of oxidative stress. As mentioned, the antioxidant activity of propolis against UV-induced skin damage has already been reported. However, considering the role of IL-6 and VEGF in promoting oxidative stress, our findings could suggest that propolis extracts might paradoxically act in a pro-oxidant manner.

For this reason, intracellular ROS production was measured in HaCaT cells challenged with H_2_O_2_ (1 mM), according to a previous experimental paradigm in which oxidative damage had been correlated with VEGF production [36]. Intriguingly, ROS levels were significantly reduced by the same concentrations of propolis extracts responsible for VEGF induction (Figure 10), thus confirming the coexistence of antioxidant effects.

### 3.7. Effect on HIF-1α Stabilization—Immunocytochemistry

As previously mentioned, the transcription factor HIF-1 is widely involved in the expression of several genes playing a role in inflammation, tissue remodeling, and wound healing. *VEGFA* is a well-known target gene downstream of HIF-1 activation [44]. Notably, other authors have investigated the role of hypoxia in triggering the expression of several inflammatory markers in HaCaT cells, reporting an increased expression of VEGF and IL-6 [45].

Therefore, we speculated on the role of green and red Brazilian propolis in the activation of HIF-1 in keratinocytes and fibroblasts, focusing on the stabilization of the HIF-1α subunit, which was investigated through fluorescent immunocytochemistry in confocal microscopy. HaCaT and HDF cells were treated in a time course for 1–3–6–24 h with green or red propolis extracts at the highest common non-toxic concentrations of 50 and 10 μg/mL, respectively, to allow for direct comparability. Dimethyloxalylglycine (DMOG) was used as a reference inducer of HIF-1α stabilization. Controls, as expected, show the absence of HIF-1α, which, in normal conditions, undergoes rapid degradation with a half-life of less than 10 min [46]. On the other hand, when cells are treated with DMOG, the activity of HIF-prolyl hydroxylases is impaired, thus leading to the accumulation of HIF-1α in the cytoplasm and promoting its prompt translocation to the nucleus, where it can be detected by immunofluorescence. At 1 and 3 h, no sign of translocation could be detected, whereas the most interesting results have been obtained at 6 and 24 h.

In HaCaT cells at 6 h (Figure 11) and 24 h (Figure 12), both green and red propolis determined a significant nuclear accumulation of HIF-1α, indicating the ability to induce HIF-1 transcriptional activity.

In HDF cells at 6 h (Figure 13), only the reference inducer DMOG caused an appreciable nuclear translocation of HIF-1α, while propolis seemed to be inactive. However, at 24 h (Figure 14), red propolis proved to be capable of inducing a significant stabilization and nuclear translocation of the transcription factor subunit, whereas green propolis remained inactive in this cell line.

## 4. Discussion

Propolis is a multifaceted bee product that has been traditionally used to treat infectious disorders, including those occurring at the skin level [2]. Its therapeutic properties, known since ancient times, are currently experiencing a renewed and deserved interest in pharmacological research. “Propolis” is an umbrella term used to define a complex product, which is only outwardly uniform in features, macroscopic appearance, and composition, but has dramatic diversity in molecular components due to the geographical area of origin, the botanical species present around the apiary, and the climatic conditions at the collection time [37]. Moreover, the methods of extraction from the raw material might also influence the final composition, leading to highly diverse propolis extracts. Despite the extreme chemical variability, which inevitably poses serious issues about product standardization, different propolis types generally share pharmacological properties in terms of antioxidant, antimicrobial, anti-inflammatory, and immunomodulatory activities [3,4,5], which constitute the primary motivation for its therapeutic exploitation. Beyond the multifariousness of propolis composition, a common thread unites samples with similar biological activities: the presence of high concentrations of flavonoids or phenolic acids. To overcome propolis’ inherent diversity and facilitate research in this field, many efforts have been made to classify and characterize propolis varieties according to their geographical origin and physical–chemical properties. Two of the varieties that are generating greater interest are green and red Brazilian propolis, which are native to diametrically opposed regions of Brazil and foraged from different botanical sources, known as *Baccharis dracunculifolia* DC. and *Dalbergia ecastaphyllum* (L.) Taub., respectively [7,8]. The most innovative field of application of these propolis varieties is probably the treatment of skin conditions: recent in vivo evidence in rodent models has, in fact, highlighted the protective effects against UV-induced oxidative stress [22,23] and the adjuvant activity in wound healing [13]. However, the modulation of inflammatory pathways at the skin level has been investigated in a few articles [47,48].

The research, to address the objective of the present work, has been conducted on multiple fronts to elucidate potential differences and strengths of the two extracts under study. A thorough chemical characterization, carried out with different and complementary chromatographic techniques, allowed the identification of numerous compounds present in green and red Brazilian propolis. The results of the analyses indicated artepillin C (8.5%, relative abundance) and drupanin (3.6%, relative abundance) as the main components of green propolis, whereas vestitol (14.6%, relative abundance), medicarpin (9.73%, relative abundance), and neovestitol (7.52%, relative abundance) were the most abundant in red propolis, thus confirming the evidence already reported in the literature [49]. This characterization confirmed the higher flavonoid content of red propolis, while in green propolis, the amount of flavonoid appears limited in favor of higher amounts of prenylated derivatives of phenolic acids. A considerable total phenolic content was demonstrated in both samples through the Folin–Ciocâlteu assay, which was significantly higher in red propolis, consistent with the work of Machado et al., who demonstrated that, among other varieties of Brazilian propolis, red propolis presented the highest phenolic content [50]. There is, in fact, a close correlation between the antioxidant capacity and the molecular structure of phenolic compounds, which was also reflected in the ORAC test (Table 2 and Table 3).

The biological activities have been investigated in two in vitro models of stable human cell lines representative of the principal actors involved in the process of wound healing, HaCaT keratinocytes and HDF dermal fibroblasts. Preliminary studies, assessing mitochondrial function and intracellular lactate dehydrogenase release, have been conducted to determine the cytotoxicity of the extracts and define treatment concentrations for subsequent experiments (Appendix A). Green propolis did not induce cytotoxic effects up to 50 μg/mL and 24 h of treatment. On the other hand, red propolis showed, in HaCaT cells, prodromal toxicity signs at the highest concentration at 6 h, which became fully manifested at 24 h, limiting the maximum usable concentration to 10 μg/mL at longer time points. According to the literature data, cytotoxic effects have been attributed to the elevated isoflavone content [51]. Nevertheless, a putative protein synthesis inhibitory mechanism has been excluded based on the results of the O-propargyl-puromycin assay.

The anti-inflammatory properties of green and red Brazilian propolis, although being quite well-established in [52], have never been explored in keratinocytes or in dermal fibroblasts. Consequently, the ability to inhibit NF-κB-driven transcription, representative of one of the principal pathways involved in skin inflammatory processes, has been extensively evaluated in HaCaT and HDF cells (Figure 4 and Figure 5). In HaCaT cells, both extracts, with red propolis in a much more significant manner, were active in inhibiting the activation of NF-κB under TNF-α pro-inflammatory stimulation. In HDF cells, the effect was less pronounced and manifested with a certain significance only for red propolis at the highest concentration tested. The ability to inhibit NF-κB-driven transcription under another pro-inflammatory stimulus, IL-1β, was further examined; however, it was only examined in HaCaT cells since this cytokine could not elicit NF-κB activation in HDF cells. These data suggested that the higher presence of polyphenols in red propolis might be responsible for both the anti-inflammatory and the cytotoxic effects observed at higher concentrations.

The NF-κB pathway is implicated in the downstream modulation of the expression of several inflammatory chemokines and cytokines, including IL-8 and IL-6. IL-8 and IL-6 are expressed in the cutis during inflammation, but an excessive or prolonged secretion is correlated with the chronicization of cutaneous injuries [53]. In line with NF-κB impairment, red propolis exerted a significant inhibitory effect on IL-8 release in both HaCaT and, to a greater extent, HDF cells under TNF-α pro-inflammatory stimulation, thus confirming the anti-inflammatory properties previously inferred (Figure 6). On the other hand, a limited or no effect was seen in the case of green propolis. Once again, the greater efficacy of red propolis emerging from these results is probably associated with the higher phenolic content and especially with the presence of flavonoids [54].

Surprisingly, both propolis samples were able to increase the expression of the *IL6* gene in basal conditions (Figure 8), paralleled by an increase in cytokine secretion levels only in the presence of a pro-inflammatory stimulus (Figure 7) but not in the absence of stimulation. This is particularly interesting considering that the favorable involvement in the initial phases of cutaneous repair of IL-6, which exerts a proliferative effect on keratinocytes, is a chemoattractant for neutrophils, and may play a role in collagen deposition and angiogenesis [42]. Notably, previous research demonstrated that *IL6* knockout mice present a significant delay in wound healing [43]. We further noticed that Brazilian green and red propolis can enhance VEGF expression in HaCaT cells (Figure 8 and Figure 9). These results are concordant with published studies ascribing wound healing properties to polyphenols featuring antioxidant effects and promoting activity on growth factors such as VEGF [55]. Moreover, they are in line with other works supporting the wound healing properties of Brazilian propolis [19,20,21]. Notably, these modulatory effects on inflammatory mediators were paralleled by the reduction in ROS levels, which, again, suggested a favorable antioxidant activity in tissue remodeling (Figure 10).

The putative molecules possibly responsible for the aforementioned anti-inflammatory activities may be recognized for red propolis in vestitol, which is able to inhibit the release of chemokines and the migration of neutrophils in the inflammatory site, and neovestitol, which modulates the NO pathway to inhibit leukocyte recruitment [56]. However, apart from this evidence, activities on the transcription factor NF-κB analogous to those demonstrated in this work have never been reported nor related to any specific component. For green propolis, the ability to modulate the NF-κB pathway, and consequently inhibit the production of prostaglandin E_2_ and nitric oxide, has been previously demonstrated for the main component artepillin C [57]. All of these molecules have been identified in the propolis samples considered in this study and, on the basis of their relative abundance, were found to be the principal characterizing compounds and, therefore, the main candidates for the attribution of the anti-inflammatory activities observed.

Polyphenols from other sources are known to interfere not only with NF-κB, but also with other pathways involved in cell metabolism, such as the HIF-1 pathway [24,25,58]. HIF-1 is a heterodimer composed of an oxygen-regulated alpha subunit, the stability of which is enhanced by hypoxia through the inhibition of prolyl hydroxylases responsible for post-translational modifications involved in ubiquitination and proteasomal degradation, and a constitutively expressed beta subunit. The HIF-1 transcription factor is able to induce the expression of genes that promote cell survival, re-establish tissue oxygenation (e.g., *VEGFA* [44]), and sustain glycolytic metabolism to produce ATP under oxygen shortage [59]. It has been recently demonstrated that the hypoxia-independent induction of HIF-1 activity, obtained with the pharmacological inhibition of HIF-prolyl hydroxylases, evokes a regenerative phenotype in mammals [60] and may therefore influence cell survival, wound closure, and tissue regeneration, suggesting that targeting this pathway can improve the wound healing process [60,61,62]. In addition, prolyl hydroxylase inhibitors such as the iron chelating agents deferoxamine and deferiprone, are successfully used for the treatment of difficult-to-heal diabetic and pressure ulcers [63,64].

Thus, the final part of our work was aimed at demonstrating the plausible involvement of HIF-1 in the mechanisms of action previously elucidated. Initial screening immunocytochemistry experiments have demonstrated that both propolis samples are able to induce the stabilization and nuclear translocation of HIF-1α in HaCaT cells, while only red propolis seems to be active at 24 h in HDF cells (Figure 11, Figure 12, Figure 13 and Figure 14). These results showed a greater activity of green propolis in HaCaT cells and reconfirmed that, in HDF cells, only red propolis is active and at longer time points. Notably, in HDF cells, the effect on HIF-1α stabilization appeared temporally disjoint from that of the reference compound DMOG, thus suggesting different underlying molecular mechanisms.

To have a better understanding of the situation, the experiments investigating *VEGFA* and *IL6* expression were repeated using DMOG as a reference compound, thus sustaining the link between HIF-1α stabilization and inflammatory modulation (Appendix A). It was noteworthy that *VEGFA* expression, but not *IL6*, was significantly increased by DMOG treatment in both cell lines, thus introducing an element of complexity, suggesting that HIF-1 could only partially explain the modulatory effects of propolis. Accordingly, DMOG showed only a mild inhibitory effect on NF-κB activation compared to red propolis extract (Appendix A). Similar evidence was reported by other authors, who demonstrated that the stabilization of the isoforms HIF-1α and HIF-2α, both active under hypoxic conditions, exacerbated the TNF-α-induced increase in VEGF and IL-6, while decreasing the levels of TSLP, despite their common dependency on the NF-κB pathway [45].

To simplify the comparison among the biological effects of red and green Brazilian propolis and DMOG, the direction of the overall effects is reported in Table 8.

## 5. Conclusions

This experimental work investigated both the phytochemical composition of two Brazilian propolis samples and their biological activity at the skin level. The complex activity on inflammatory and proliferative mediators (IL-8, IL-6, and VEGF), observed in human keratinocytes and fibroblasts, is correlated with the divergent modulation of the transcription factors NF-κB and HIF-1. Plausibly, bioactive concentrations considered in the study (10–50 μg/mL) may be achieved after topical application of milligrams of propolis in animals and human studies. Collected data may contribute to strengthening the biological rationale behind the traditional use of Brazilian propolis in inflammatory disorders associated with skin lesions. More specifically, the biological properties of Brazilian propolis may influence skin repair through immunomodulatory mechanisms in addition to classical antimicrobial effects. These results may inform the design of future studies, in which the range of concentrations used in the present work would be easily translated to address in vivo safety and efficacy.

## Figures and Tables

**Figure 1 biomedicines-13-02229-f001:**
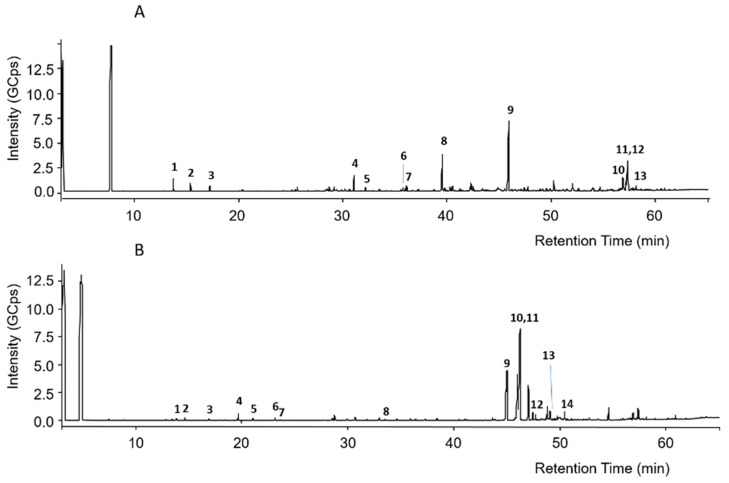
GC-MS chromatograms of silylated green propolis (**A**) and red propolis (**B**). Numbers indicate the principal peaks that were putatively assigned a structure through a comparison of mass spectra with those found in the literature.

**Figure 2 biomedicines-13-02229-f002:**
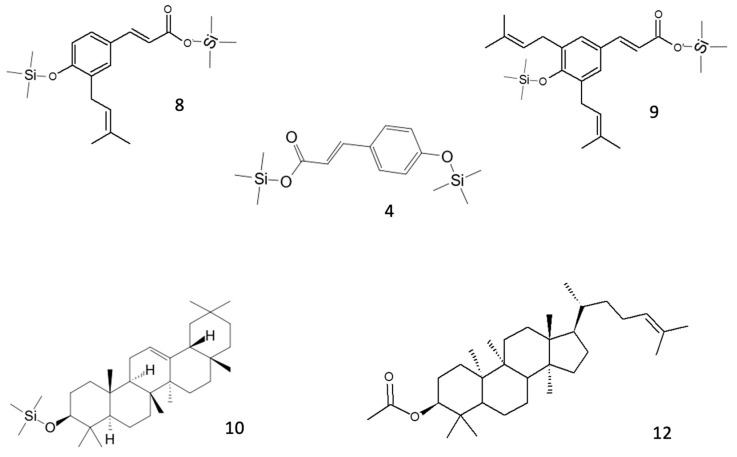
Structures of the main putative compounds identified in green propolis through GC-EI-MS: *p*-coumaric acid (**4**), drupanin 2TMS (**8**), artepillin C 2TMS (**9**), β-amyrin TMS (**10**), and cycloartenol acetate (**12**).

**Figure 3 biomedicines-13-02229-f003:**
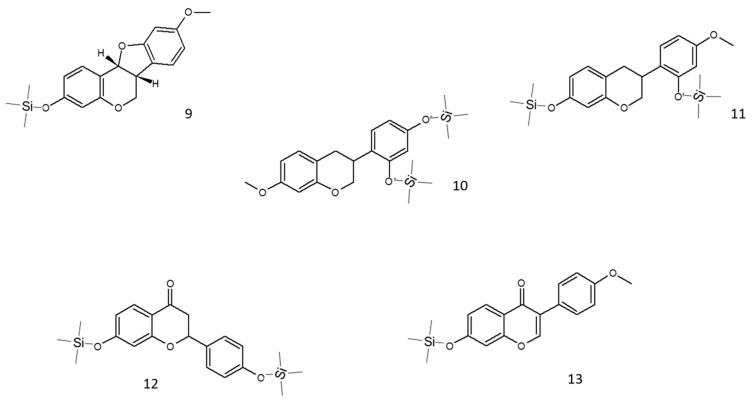
Structures of the main putative compounds identified in red propolis through GC-EI-MS: medicarpin TMS (**9**), neovestitol 2TMS (**10**), vestitol 2TMS (**11**), liquiritigenin 2TMS (**12**), and formononetin TMS (**13**).

**Figure 4 biomedicines-13-02229-f004:**
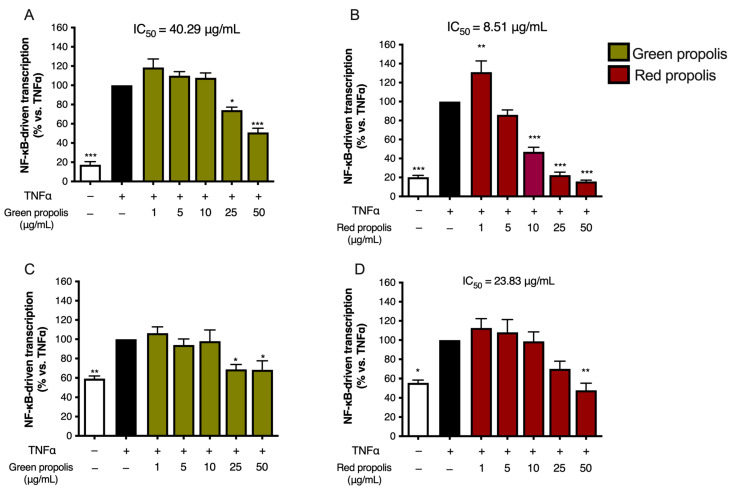
Assessment of green (**A**,**C**) and red (**B**,**D**) Brazilian propolis effect on NF-κB-driven transcription in HaCaT (**A**,**B**) and HDF (**C**,**D**) cells. Cells were stimulated with 10 ng/mL of TNF-α and treated for 6 h with increasing propolis concentrations. EGCG 20 μM was used as a reference inhibitor (−77%). Data are expressed as a percentage of the stimulus, which was arbitrarily assigned the value of 100%. − means “present” and + means “absent”. * *p* < 0.05, ** *p* < 0.01, and *** *p* < 0.001 versus TNF-α.

**Figure 5 biomedicines-13-02229-f005:**
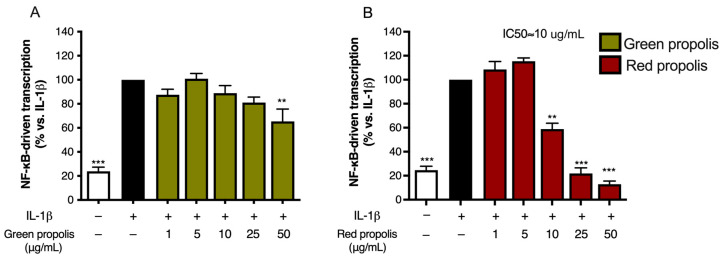
Assessment of green (**A**) and red (**B**) Brazilian propolis effect on NF-κB-driven transcription in HaCaT cells. Cells were stimulated with 10 ng/mL of IL-1β and treated for 6 h with increasing propolis concentrations. DMOG 1 mM was used as a reference inhibitor (−41%). Data are expressed as a percentage of the stimulus, which was arbitrarily assigned the value of 100%. − means “present” and + means “absent”. ** *p* < 0.01 and *** *p* < 0.001 versus IL-1β.

**Figure 6 biomedicines-13-02229-f006:**
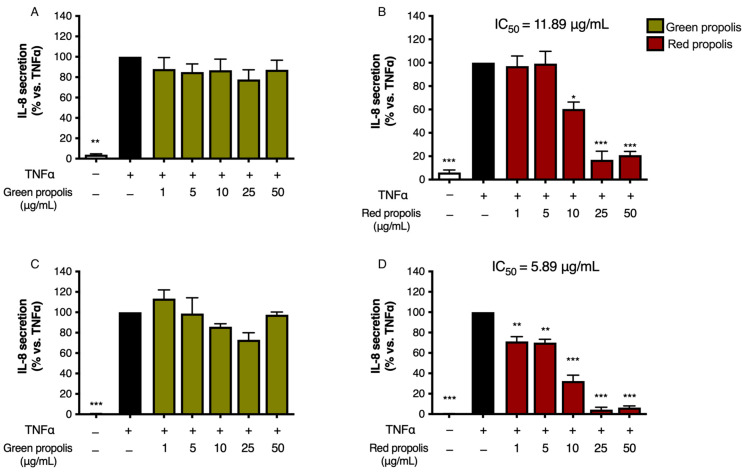
Assessment of green (**A**,**C**) and red (**B**,**D**) Brazilian propolis effect on IL-8 release in HaCaT (**A**,**B**) and HDF (**C**,**D**) cells. Cells were stimulated with 10 ng/mL of TNF-α and treated for 6 h with increasing propolis concentrations. EGCG 20 μM was used as a reference inhibitor (−75%). Data are expressed as a percentage of the stimulus, which was arbitrarily assigned the value of 100%. − means “present” and + means “absent”. * *p* < 0.05, ** *p* < 0.01, and *** *p* < 0.001 versus TNF-α.

**Figure 7 biomedicines-13-02229-f007:**
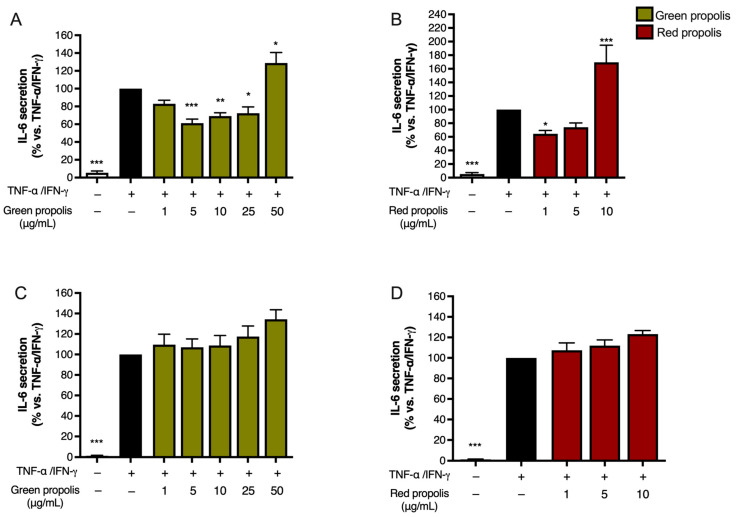
Assessment of green (**A**,**C**) and red (**B**,**D**) Brazilian propolis effect on IL-6 release in HaCaT (**A**,**B**) and HDF (**C**,**D**) cells. Cells were stimulated with 10 ng/mL of TNF-α and 5 ng/mL of IFN-γ and treated for 24 h with increasing propolis concentrations. EGCG 20 μM was used as a reference inhibitor (−79%). Data are expressed as a percentage of the stimulus, which was arbitrarily assigned the value of 100%. − means “present” and + means “absent”. * *p* < 0.05, ** *p* < 0.01, and *** *p* < 0.001 versus TNF-α/IFN-γ.

**Figure 8 biomedicines-13-02229-f008:**
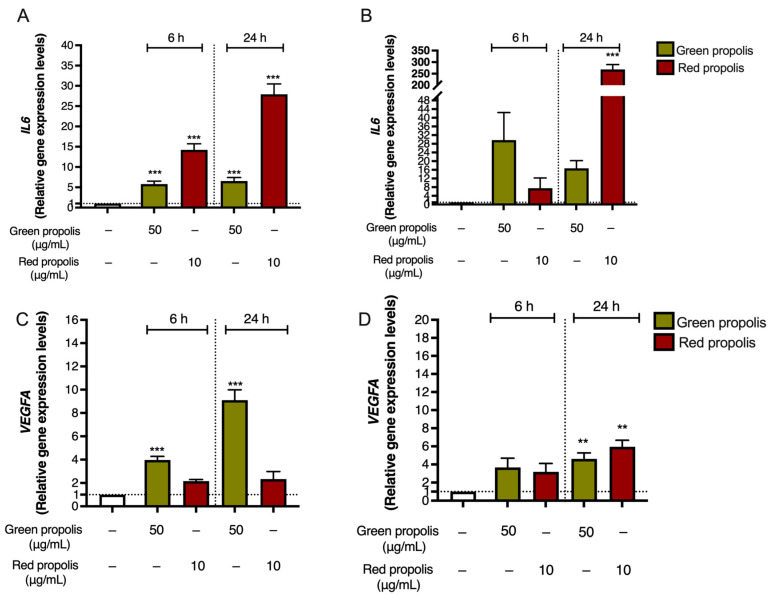
Gene expression analysis of *IL6* (**A**,**B**) and *VEGFA* (**C**,**D**) in HaCaT (**A**,**C**) and HDF (**B**,**D**) cells treated for 6 or 24 h with green and red Brazilian propolis. − means “present”. Dotted line means “cut-off/reference” ** *p* < 0.01 and *** *p* < 0.001 versus control (white bar).

**Figure 9 biomedicines-13-02229-f009:**
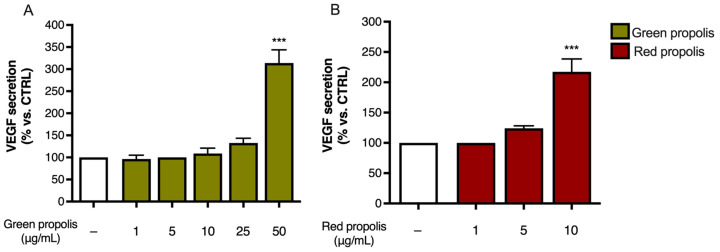
Assessment of green (**A**) and red (**B**) Brazilian propolis effect on VEGF release in HaCaT cells. Cells were treated for 24 h with increasing propolis concentrations. Data is expressed as a percentage of the control, which was arbitrarily assigned the value of 100%. − means “present”. *** *p* < 0.001 vs. control (white bar).

**Figure 10 biomedicines-13-02229-f010:**
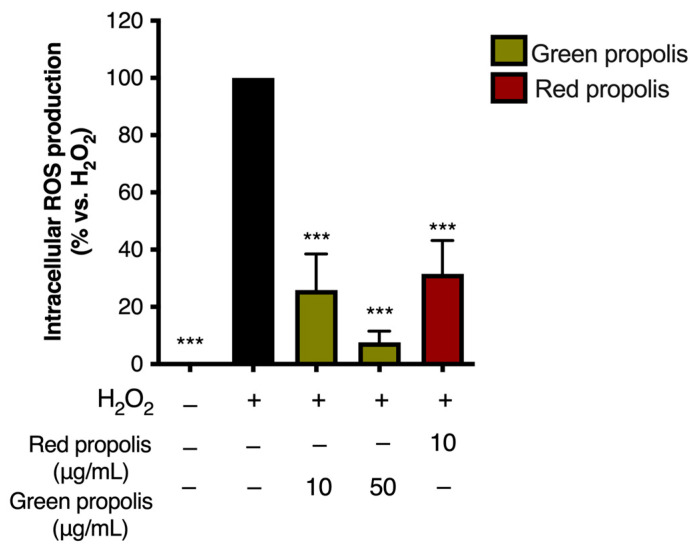
Assessment of green and red Brazilian propolis effect on ROS production in HaCaT cells. Cells were treated for 24 h with propolis extracts; then, cells were challenged with a pro-oxidant stimulus (H_2_O_2_, 1 mM) for 1 h. Data are expressed as a percentage of the stimulus, which was arbitrarily assigned the value of 100%. − means “present” and + means “absent”. *** *p* < 0.001 vs. H_2_O_2_.

**Figure 11 biomedicines-13-02229-f011:**
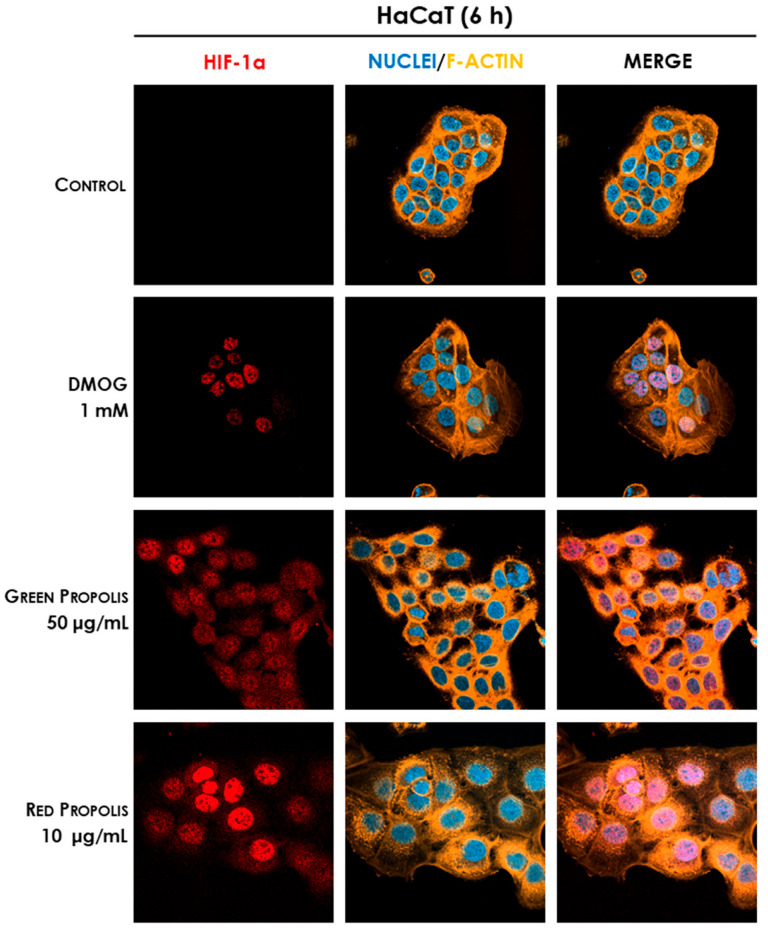
Representative confocal micrographs of the fluorescent immunostaining of HIF-1α (red) to assess the effect of Brazilian propolis on HIF-1α stabilization and nuclear translocation in HaCaT cells after 6 h of treatment (magnification 60×, 50 μm scale). Nuclei were stained with DAPI (blue), whereas F-actin was stained with TRITC-phalloidin (orange).

**Figure 12 biomedicines-13-02229-f012:**
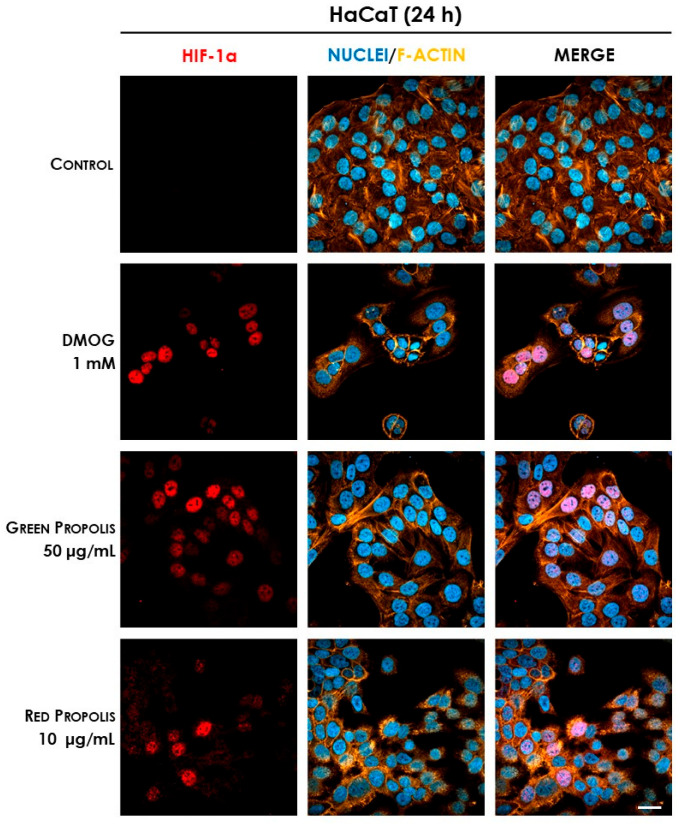
Representative confocal micrographs of the fluorescent immunostaining of HIF-1α (red) to assess the effect of Brazilian propolis on HIF-1α stabilization and nuclear translocation in HaCaT cells after 24 h of treatment (magnification 60×, 50 μm scale). Nuclei were stained with DAPI (blue), whereas F-actin was stained with TRITC-phalloidin (orange).

**Figure 13 biomedicines-13-02229-f013:**
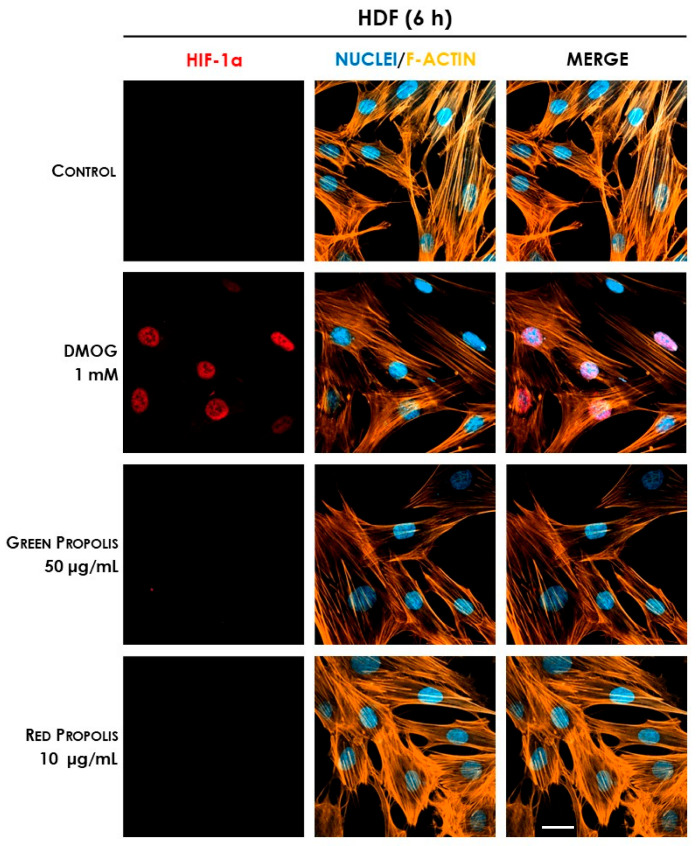
Representative confocal micrographs of the fluorescent immunostaining of HIF-1α (red) to assess the effect of Brazilian propolis on HIF-1α stabilization and nuclear translocation in HDF cells after 6 h of treatment (magnification 60×, 50 μm scale). Nuclei were stained with DAPI (blue), whereas F-actin was stained with TRITC-phalloidin (orange).

**Figure 14 biomedicines-13-02229-f014:**
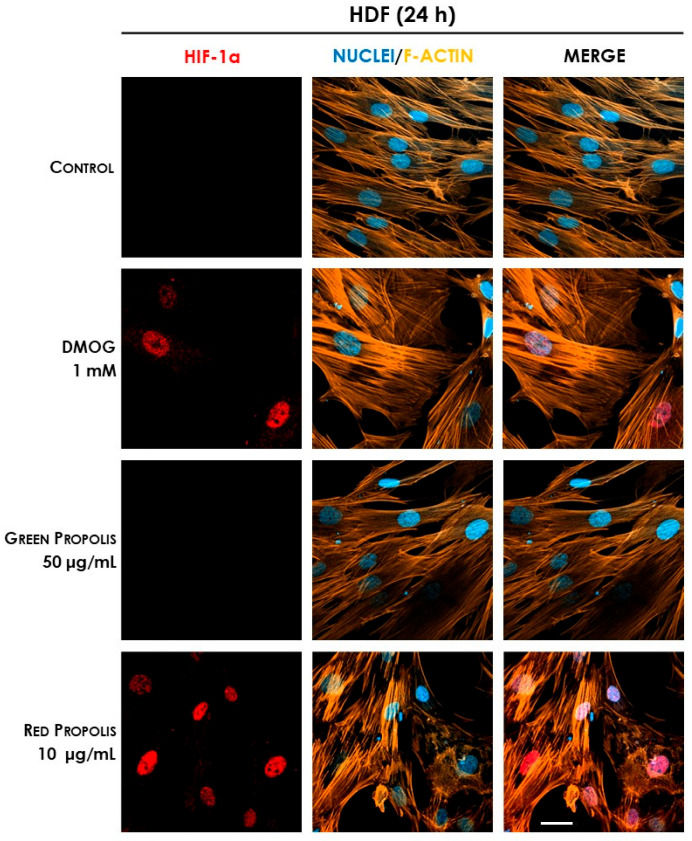
Representative confocal micrographs of the fluorescent immunostaining of HIF-1α (red) to assess the effect of Brazilian propolis on HIF-1α stabilization and nuclear translocation in HDF cells after 24 h of treatment (magnification 60×, 50 μm scale). Nuclei were stained with DAPI (blue), whereas F-actin was stained with TRITC-phalloidin (orange).

**Table 1 biomedicines-13-02229-t001:** Sequence of primers used in this study.

Gene		5′→3’ Sequence
h*VEGFA*	Forward	CGAGGCAGCTTGAGTTAA
	Reverse	CTGTATCAGTCTTTCCTGGTG
	Probe	CTCGGCTTGTCACATCTGCAAGT
h*IL6*	Forward	GGGAACGAAAGAGAAGCTC
	Reverse	AGGCAACTGGACCGAA
	Probe	CGCTTGTGGAGAAGGAGTTCAT
h*36B4*	Forward	CCACGCTGCTGAACATGC
	Reverse	TCGAACACCTGCTGGATGAC
	Probe	AACATCTCCCCCTTCTCCTTTGGGCT

**Table 2 biomedicines-13-02229-t002:** Total phenolic content (TPC) of Brazilian green and red propolis extracts. The results are expressed as mg of gallic acid per g of extract or tincture (mean ± SD).

	TPC (mg GAE/g Dry Extract)	TPC (mg GAE/g Hydroethanolic Tincture)
Green propolis	100.67 ± 7.94	11.63 ± 0.92
Red propolis	200.59 ± 12.80	17.84 ± 1.14
*p*-value	0.0003	0.0018

GAE: gallic acid equivalent; *p*-value (*t*-test, green vs. red propolis).

**Table 3 biomedicines-13-02229-t003:** Oxygen Radical Absorbance Capacity (ORAC) values of Brazilian green and red propolis extracts. The results are expressed as mmol of Trolox per g of extract or tincture (mean ± SD).

	ORAC Value (mmol Trolox eq./g Dry Extract)	ORAC Value (mmol Trolox eq./g Hydroethanolic Tincture)
Green propolis	44.511 ± 3.659	5.144 ± 0.423
Red propolis	98.009 ± 0.019	8.719 ± 0.002
*p*-value	0.0023	0.0069

*p*-value (*t*-test, green vs. red propolis).

**Table 4 biomedicines-13-02229-t004:** Putatively annotated compounds identified by GC-EI-MS in green propolis.

Peak Number	RT (min)	Compound	Relative %
1	13.763	Glycerol 3TMS	0.9
2	15.411	Di-hydrocinnamic acid, ethyl ester	0.6
3	17.261	Di-hydrocinnamic acid TMS	0.4
4	31.095	*p*-coumaric acid 2TMS	1.3
5	32.202	Palmitic acid, ethyl ester	0.3
6	35.791	Caffeic acid 3TMS	0.3
7	36.137	Oleic acid, ethyl ester	0.5
8	39.541	Drupanin 2TMS	3.6
9	45.923	Artepillin C 2TMS	8.5
10	56.864	β-amyrin TMS	1.8
11	57.204	α-amyrin TMS	1.0
12	57.310	Cycloartenol acetate	3.4
13	58.115	Lupeol acetate	0.4

TMS: trimethylsilyl group; RT: retention time.

**Table 5 biomedicines-13-02229-t005:** Putatively annotated compounds identified by GC-EI-MS in red propolis. TMS: trimethylsilyl group.

Peak Number	RT (min)	Compound	Relative %
1	13.794	Estragole	0.13
2	14.591	Succinic acid 2TMS	0.24
3	16.861	Methyleugenol	0.11
4	19.627	Malic acid 3TMS	0.57
5	21.015	Elemicin	0.17
6	23.099	Eugenol TMS	0.21
7	23.612	Asarone	0.07
8	33.498	Palmitic acid TMS	0.09
9	44.993	Medicarpin TMS	9.73
10	45.963	Neovestitol 2TMS	7.52
11	46.205	Vestitol 2TMS	14.60
12	47.429	Liquiritigenin 2TMS	0.8
13	49.05	Formononetin TMS	1.12
14	50.422	Isoliquiritigenin 3TMS	0.94

**Table 6 biomedicines-13-02229-t006:** Compounds identified in green propolis through HPLC-ESI-HRMS analysis.

Peak Number	RT (min, UV/PDA)	RT (min, MS-TOF)	*m/z* exp. ESI (−)	*m/z* calc (−)	Compound	λ_max_	Formula
1	7.93	7.959	163.0399	163.0401	*p*-coumaric acid ^§^	228, 310	C_9_H_8_O_3_
2	12.40	12.457	515.1195	515.1195	Di-O-caffeoylquinic acid	244, 327	C_25_H_24_O_12_
3	13.54	13.605	515.1190	515.1195	Di-O-caffeoylquinic acid	244, 326	C_25_H_24_O_12_
4	25.73	25.724	231.1029	231.1027	Drupanin ^§^	236, 315	C_14_H_16_O_3_
5	27.54	27.607	315.1601	-	Not identified	240, 314	C_19_H_23_O_4_
6	29.93	30.019	389.1965	299.0561	Kaempferide	266, 369	C_16_H_12_O_6_
7	30.62	30.684	329.0660	329.0667	Dimethylquercetin	370	C_17_H_14_O_7_
8	39.42	39.548	299.1633	299.1653	Artepillin C ^§^	238, 314	C_19_H_24_O_3_
9	45.60	45.591	363.1591	363.1602	Baccharin	283	C_23_H_24_O_4_

^§^ compounds identified in GC-MS analysis as well.

**Table 7 biomedicines-13-02229-t007:** Compounds identified in red propolis through HPLC-ESI-HRMS analysis.

Peak Number	RT (min, UV/PDA)	RT (min, MS-TOF)	*m/z* exp. ESI (−)	*m/z* calc (−)	Compound	λ_max_	Formula
1	15.30	15.349	255.0657	255.2460	Liquiritigenin ^§^/ isoliquiritigenin ^§^	236, 276	C_15_H_12_O_4_
2	23.66	23.701	267.0658	267.2567	Formononetin ^§^/ isoformononetin	249	C_16_H_11_O_4_
3	24.23	24.284	271.0974	271.2884	Vestitol/neovesitol ^§^	202, 280	C_16_H_16_O_4_
4	25.99	26.031	271.0972	271.2884	Vestitol/neovesitol ^§^	202, 280	C_16_H_16_O_4_
5	26.76	26.801	269.0813	269.2726	•Medicarpin ^§^ •4,4′-dihydroxi-2-methoxychalcone •(7 S)-dalbergiphenol •Pinostrobin	205, 287	C_16_H_14_O_4_ C_16_H_14_O_4_ C_16_H_14_O_4_ C_16_H_14_O_4_
6	54.06	54.148	601.3535	601.7926	Guttiferone	252, 354	C_38_H_50_O_6_
7	54.21	54.303	601.3531	601.7926	Guttiferone	252, 354	C_38_H_50_O_6_
8	54.79	54.88	501.3003	501.6768	Nemorosone	305	C_33_H_42_O_4_

^§^ compounds identified in GC-MS analysis as well.

**Table 8 biomedicines-13-02229-t008:** Comparison between the biological effects of Brazilian propolis and HIF-1α stabilizer DMOG, demonstrated in HaCaT and HDF cells.

	HIF-1α Stabilizer	Brazilian Propolis			
	DMOG	Red	?	Green	?
IL-8	⇓	⇓	Yes	n.a.	No
IL-6	n.a.	⇑	No	⇑	No
VEGF	⇑	⇑	Yes	⇑	Yes
NF-κB	⇓	⇓	Yes	⇓	Yes
HIF-1α	⇑	⇑	Yes	⇑	Yes

⇓ reduction, ⇑ increase; n.a., not active; ?, the activity of Brazilian propolis resembles HIF-1α stabilization (yes/no).

## Data Availability

Data are available upon request to the corresponding author (stefano.piazza@unimi.it).

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
