# Peer review of "Assessment of the Polyphenolic Profile and Beneficial Effects of Red and Green Propolis in Skin Inflammatory Conditions and Oxidative Stress"

_biomedicines, 2025, doi:10.3390/biomedicines13092229_

Round 1

Reviewer 1 Report

Comments and Suggestions for Authors

The manuscript entitled "Assessment of the Polyphenolic Profile and Beneficial Effects of Red and Green Propolis in Skin Inflammatory Conditions and Oxidative Stress" presents an interesting investigation into the biological mechanisms of action of propolis, which is a valuable contribution to the field.

However, the manuscript is excessively long and includes numerous chemical profiling protocols that could be streamlined. Many of the chemical standards used are already well established and commercially avaiable. In particular, the HPLC fingerprint for green propolis has been available for a long time, and more recently, the chemical profiles for both red and green propolis were published by the International Organization for Standardization (ISO).

If the authors chose to combine three chemical techniques for their evaluation, this is certainly within their discretion. However, despite this effort, the manuscript lacks quantification of the principal constituents in each type of propolis. Given the existing literature and the availability of reference standards, it would have been feasible to perform both fingerprinting and accurate quantification using a conventional HPLC-DAD-UV method. Typically, 280 nm is used as the detection wavelength for propolis analysis. Is it necessary to present chromatograms at both 325 nm and 280 nm? My suggestion is to revise the chemical characterization reducing what is possible since this is already well described.

Although the manuscript presents HPLC-DAD-UV results, the corresponding methodology is not described in the methods section. This omission makes it difficult to compare the obtained fingerprints with those in previously published studies. Please ensure that the method is fully described and cite the appropriate scientific references.

The introduction is overly long and contains information that is not directly relevant to the main objectives of the study. While the content in lines 42–53 is interesting, it does not contribute to the core purpose of the manuscript and may be omitted. The authors may consider this suggestion. Additionally, in Brazil, propolis is primarily produced by Africanized Apis mellifera, a hybrid species. Therefore, the phrase “western honey bees” in line 38 should be corrected to “Africanized honey bees.”

Is the content in lines 95–97 necessary?

Similarly, the content in lines 119–122 (“based on... evaluated”) should be reconsidered for its relevance.

Given that the authors selected HaCaT keratinocytes and HDF dermal fibroblasts—two cell types with distinct physiological behaviors—it would be beneficial to briefly discuss the physiological and pharmacological roles of each in the wound healing process. This context could help clarify the differences observed between treatments and strengthen the discussion.

Considering the observed differences in potency and behavior between red and green propolis, have the authors considered the possibility of formulating a synergistic combination of both types? Furthermore, in light of the manuscript's potential application in topical product development, it would be valuable for the authors to propose a conversion of the in vitro dosage to an estimated in vivo dosage. The manuscript clearly demonstrates that red propolis exhibited toxicity at lower concentrations than green propolis. The authors’ opinion on this matter would enhance the practical relevance of the study and broaden its implications for both academic and industrial applications.

Author Response

We are pleased to receive the comments of the reviewer, which were appreciated and improved our manuscript. Point-by-point answers are reported below:

The manuscript entitled "Assessment of the Polyphenolic Profile and Beneficial Effects of Red and Green Propolis in Skin Inflammatory Conditions and Oxidative Stress" presents an interesting investigation into the biological mechanisms of action of propolis, which is a valuable contribution to the field.

However, the manuscript is excessively long and includes numerous chemical profiling protocols that could be streamlined. Many of the chemical standards used are already well established and commercially avaiable. In particular, the HPLC fingerprint for green propolis has been available for a long time, and more recently, the chemical profiles for both red and green propolis were published by the International Organization for Standardization (ISO). If the authors chose to combine three chemical techniques for their evaluation, this is certainly within their discretion. However, despite this effort, the manuscript lacks quantification of the principal constituents in each type of propolis. Given the existing literature and the availability of reference standards, it would have been feasible to perform both fingerprinting and accurate quantification using a conventional HPLC-DAD-UV method. Typically, 280 nm is used as the detection wavelength for propolis analysis. Is it necessary to present chromatograms at both 325 nm and 280 nm? My suggestion is to revise the chemical characterization reducing what is possible since this is already well described. Although the manuscript presents HPLC-DAD-UV results, the corresponding methodology is not described in the methods section. This omission makes it difficult to compare the obtained fingerprints with those in previously published studies. Please ensure that the method is fully described and cite the appropriate scientific references.

We thank the reviewer for useful suggestions. LC-MS and GC-MS were maintained, while HPLC-DAD-UV analysis was removed by the main text. This choice reflects the observations made by the reviewer regarding the availability of HPLC-DAD-UV fingerprint in the literature. In addition, LC-MS and GC-MS data are in line with chemical profiles reported by ISO 24381:2023. Despite our chemical analysis was conducted before 2023, thus ISO 24381:2023 were still not available, a sentence and reference about ISO 24381:2023(E) was also included in the text (line 52).

The introduction is overly long and contains information that is not directly relevant to the main objectives of the study. While the content in lines 42–53 is interesting, it does not contribute to the core purpose of the manuscript and may be omitted.

Introduction was shortened by removing lines 42-53, accordingly.

The authors may consider this suggestion. Additionally, in Brazil, propolis is primarily produced by Africanized Apis mellifera, a hybrid species. Therefore, the phrase “western honey bees” in line 38 should be corrected to “Africanized honey bees.”

The sentence was a general reference to propolis, regardless its origin, so in this context other propolis are included. Following the suggestion, “western” has been removed to avoid misunderstandings.

Is the content in lines 95–97 necessary?

Lines 95-97 have been removed.

Similarly, the content in lines 119–122 (“based on... evaluated”) should be reconsidered for its relevance.

The sentence has been removed.

Given that the authors selected HaCaT keratinocytes and HDF dermal fibroblasts—two cell types with distinct physiological behaviors—it would be beneficial to briefly discuss the physiological and pharmacological roles of each in the wound healing process. This context could help clarify the differences observed between treatments and strengthen the discussion.

We thank the reviewer for suggestions improving the introduction: a sentence was added (line 94) to summarize the role of keratinocytes and dermal fibroblasts in skin repair.

Considering the observed differences in potency and behavior between red and green propolis, have the authors considered the possibility of formulating a synergistic combination of both types?  

We thank the reviewer for the suggestion. The main focus of this work was to attribute a potential mechanism of wound healing to each Brazilian propolis, which was a missing information in literature. Considering that the two different types of propolis showed comparable biological effects, it was our opinion that the synergism among them might represent a less promising aspect, although of interest. Moreover, it represented a further element of complexity in our experimental study. However, the suggestion might be taken into consideration for future investigation.

Furthermore, in light of the manuscript's potential application in topical product development, it would be valuable for the authors to propose a conversion of the in vitro dosage to an estimated in vivo dosage. The manuscript clearly demonstrates that red propolis exhibited toxicity at lower concentrations than green propolis. The authors’ opinion on this matter would enhance the practical relevance of the study and broaden its implications for both academic and industrial applications.

We thank the reviewer for this suggestion: two sentences regarding the potential relevance of our results for human and vivo data were added in the “conclusions” section. Considering a minimum of 1-5% delivery into viable skin, we might estimate that our treatments, conducted at 10 ug/mL concentration, might translate in a dose range of mg by topical application.

Reviewer 2 Report

Comments and Suggestions for Authors

The author in the manuscript “Assessment of the polyphenolic profile and beneficial effects of red and green propolis in skin inflammatory conditions and oxidative stress” is an interesting study and could be considered for publication in the journal “Pharmaceuticals”. However, there are certain points to be addressed to improve the manuscript.

  1. L-20: the word technics should be replaced with technique.
  2. L-163: The heading should be GC-MS analysis
  3. L-164-166: Authors are suggested to briefly describe the chemistry of this reaction and why it was carried out.
  4. L-358: It will be more authentic if the authors run standard phenolics. If possible they should run the standard of identified phenolics.
  5. What is the concentrations of phenolics in green and red propolis in equal quantity of the samples? Quantitative data will clearly explain the roadmap of inflammatory biomarkers.

Author Response

The author in the manuscript “Assessment of the polyphenolic profile and beneficial effects of red and green propolis in skin inflammatory conditions and oxidative stress” is an interesting study and could be considered for publication in the journal “Pharmaceuticals”.

The authors thank the reviewer for appreciation.

However, there are certain points to be addressed to improve the manuscript.

  1. L-20: the word technics should be replaced with technique.
  2. L-163: The heading should be GC-MS analysis

We thank the reviewer for corrections, point 1 and 2 have been addressed.

  1. L-164-166: Authors are suggested to briefly describe the chemistry of this reaction and why it was carried out.

The reaction was only briefly described for conciseness; however, a reference of this widespread method was reported in the main text (34).

  1. L-358: It will be more authentic if the authors run standard phenolics. If possible they should run the standard of identified phenolics.

The decision to use Trolox as a reference antioxidant compound was taken for a better comparison with the available literature, in which Trolox is used as reference standard. Although of interest, the comparison with individual polyphenols might not allow comparable data.

  1. What is the concentrations of phenolics in green and red propolis in equal quantity of the samples? Quantitative data will clearly explain the roadmap of inflammatory biomarkers.

We thank the reviewer for this important question. As reported in the discussion, our work correlated the higher phenolic content with more promising anti-inflammatory effects. At the same time, cytotoxicity was observed for red propolis at lower concentrations. These results suggested a deeper investigation into bioactive compounds devoid of toxic effects, including their relationship with the whole propolis extract. Actually, we are still working on the attribution of the biological effects to specific compounds through bio-guided fractionation, since, in our opinion, it remains a still open question. This is the main reason why we demanded the quantification of individual compounds.

Reviewer 3 Report

Comments and Suggestions for Authors

The manuscript presents the characterization of the polyphenol profile of two green and red Brazilian propolis extracts as well as their beneficial anti-inflammatory effects on the skin and against oxidative stress.

It is a complete and complex manuscript, but a little unbalanced. It presents in great detail the characterization of the structural composition of the two extracts to the detriment of the biological activity part. So in a first phase I would suggest reducing this part because many data have been reported in other similar articles.

What would be the more accurate explanation for why green propolis extract has lower cytotoxicity compared to red propolis extract?

Why are the studies conducted to demonstrate anti-inflammatory and anti-oxidative stress action not specifying the reference substances with whose results the tested extracts were compared?

Can it be established in the case of these two extracts which of the ROS are inhibited and due to which oxidative or hydrolytic mechanism?

Author Response

The manuscript presents the characterization of the polyphenol profile of two green and red Brazilian propolis extracts as well as their beneficial anti-inflammatory effects on the skin and against oxidative stress.

It is a complete and complex manuscript, but a little unbalanced. It presents in great detail the characterization of the structural composition of the two extracts to the detriment of the biological activity part. So in a first phase I would suggest reducing this part because many data have been reported in other similar articles.

The authors thank the reviewer for comments and agree with the observation of the reviewer: the analytical part of our investigation was reduced to allow for a better understanding of the biological data.

What would be the more accurate explanation for why green propolis extract has lower cytotoxicity compared to red propolis extract?

The authors thank the reviewer for this important question. We considered this point, at least partially, in the discussion (line 724, line 738): our work correlated the higher phenolic content of red propolis with more promising anti-inflammatory effects. Plausibly, cytotoxicity might be correlated with compounds also belonging to polyphenols. According to the question raised by the reviewer, the discussion was extended by adding the sentence at line 751. By logic conclusion, we supposed that green propolis had lower cytotoxicity due to lower polyphenolic content. However, for the knowledge of the reviewer, we are still working on the attribution of cytotoxicity and biological activity to specific compounds by performing a bio-guided fractionation of propolis extracts.

Why are the studies conducted to demonstrate anti-inflammatory and anti-oxidative stress action not specifying the reference substances with whose results the tested extracts were compared?

We apologize for omitting reference inhibitors used in biological assays: EGCG 20 uM and DMOG 1mM were used as antioxidant and anti-inflammatory reference compounds. A sentence was added to methods (“cell treatment”), and the degree of inhibition was reported in each figure caption.

Can it be established in the case of these two extracts which of the ROS are inhibited and due to which oxidative or hydrolytic mechanism?

Our antioxidant measurements capture global redox-quenching capacity rather than individual ROS species; therefore, from the present data we cannot ascribe inhibition to a specific ROS (e.g., superoxide, hydrogen peroxide, hydroxyl radical) or to a defined enzymatic mechanism. However, multiple mechanisms might be responsible for the observed effect: in this work we suggested a scavenging activity and PHD modulation. ORAC was run with AAPH, which generates alkyl peroxyl radicals; both extracts scavenged these. In HaCaT cells pretreated with the extracts and then challenged with Hâ‚‚Oâ‚‚, intracellular oxidants measured by CM-Hâ‚‚DCFDA were reduced. This probe is non-selective (it reports a general oxidative burden), so we cannot describe whether superoxide, hydroxyl radical, or Hâ‚‚Oâ‚‚ itself was preferentially affected. The extracts modulated NF-κB and stabilized HIF-1α; those results are compatible with indirect redox effects (e.g., phenolic metal chelation; possible PHD modulation upstream of HIF-1). The observed effects are consistent with polyphenol-mediated electron-transfer/hydrogen-atom transfer chemistry and possible upstream modulation of cellular antioxidant defenses, but species-specific or enzyme-level mechanisms would require targeted assays (e.g., superoxide, Hâ‚‚Oâ‚‚, and hydroxyl-specific probes; NOX/MPO/LOX activity; hyaluronidase/elastase inhibition).

Round 2

Reviewer 2 Report

Comments and Suggestions for Authors

The authors have addressed the comments satisfactorily.